

# Efficient sequestration of lead from aqueous systems by peanut shells and compost: evidence from fixed bed column and batch scale studies

Ghulam Mustafa Shah, Muhammad Imran, Umme Aiman, Muhammad Mohsin Iqbal, Muhammad Akram, Hafiz Muhammad Rashad Javeed, Atika Waqar and Faiz Rabbani

Environmental Sciences, COMSATS University Islamabad, Vehari, Punjab, Pakistan

## ABSTRACT

Lead (Pb) is a pervasive contaminant and poses a serious threat to living beings. The present study aims at batch and fixed bed column scale potential of commercial compost (CCB) and peanut shells biosorbents (PSB) for the sequestration of Pb from contaminated aqueous systems. The PSB and CCB were characterized with FTIR, SEM and Brunauer Emmett-Teller (BET) to get insight of the adsorption behavior of both materials. Fixed bed column scale experiments were performed at steady state flow (2.5 and 5.0 mL/min), initial Pb concentrations (25 and 50 mg/L) and dosage of each adsorbent (3.0 and 6.0 g/column). Columns packed (15.9 cm$^2$) with PSB and CCB have revealed excellent adsorption of Pb with PSB as compared with CCB. The total volume of injected contaminated water was 1,500 mL and 3,000 mL at 2.5 and 5.0 mL/min, respectively while total bed volume number was 157. A series of batch experiments with CCB and PSB was conducted at adsorbent dosage (1.25–5.0 g/L), initial Pb level (25–100 mg/L), interaction time (0–180 min) and solution pH (4–10) at room temperature. Batch scale results revealed that PSB removed 92% Pb from water at 25 mg Pb/L concentration as compared with CCB (79%). The presence of competing ions in groundwater showed less Pb removal as compared with synthetic water. The experimental data were simulated with equilibrium isothermal models: Langmuir, Freundlich, and kinetic models: pseudo first order, pseudo second order and intra-particle diffusion. The Freundlich and pseudo second order models better described the equilibrium and kinetic experimental data, respectively with maximum sorption of 42.5 mg/g by PSB which is also evident from FTIR functional groups and SEM results. While equilibrium sorption of Pb onto CCB was equally explained by Freundlich and Langmuir models. These findings indicate that PSB could be an active and ecofriendly biosorbent for the sequestration of metals from contaminated aqueous systems.

Corresponding author
Muhammad Imran,
imranrb@ciitvehari.edu.pk

## INTRODUCTION

Water is very crucial for all the living organisms and is a basic requirement of life. Safe drinking water is essential for sustaining life (*Shahid et al., 2021*; *Natasha et al., 2021*). With the passage of time, many natural and anthropogenic activities are polluting freshwater bodies. Several organic and inorganic pollutants are continuously discharged into surface and subsurface water bodies, thereby adversely affecting the ecosystem. In comparison of these water pollutants, heavy metal (HMs) contamination is a major concern. Heavy metals are known as dangerous metals having 63.5–200.6 g atomic weight and five times higher density than water (*Izah, Bassey & Ohimain, 2018*). Because of their toxicity to plants, animals, and humans, HMs are categorized under environmental pollutants. Heavy metals like Lead, Cadmium, Chromium, Mercury, Arsenic, and Nickel cause a serious threat to human health. These HMs are highly soluble and enter the food chain in variety of ways. These are non-degradable and persistence in nature. Natural (weathering of rocks, volcanos, floods *etc.*) and human activities (mining, petroleum refining, tanneries, battery production, metal plating, glass production and pesticides production *etc.*) induce accumulation of these HMs into the environment (*Kadirvelu, Thamaraiselvi & Namasivayam, 2001*).

Lead (Pb) is an abundant pollutant in the environment and due to its implications it becomes a major concern worldwide. In America, Pb is ranked 2nd position among all known hazardous substances (*ATSDR, 2007*). Some sources of Pb in wastewater are effluents from smelting, tanneries, metal plating, radiator manufacturing, alloy, and battery industries (*Kadirvelu, Thamaraiselvi & Namasivayam, 2001*). In drinking water, acceptable level of Pb is 0.05 mg/L set by WHO and EPA (*Arbabi, Hemati & Amiri, 2015*). Moreover, effluents from manufacturing process such as pigments, television tubes, fuel, paints, explosives are also causes of the water contamination with Pb (*Ahalya, Kanamadi & Ramachandra, 2005*). When this Pb contaminated water interacts with ecosystem, the metal ions can accumulate into human body through the consumption of Pb contaminated food chain or water (*Abdel-Salam, 2018*). A small amount of Pb continuously accumulating for extended period results in chronic toxicity and damages to organs (*Badmus, Audu & Anyata, 2007*). Moreover, Pb causes kidney damage, reproductive system, and central nervous system (CNS) in human and causes oxidative stress, damages photosynthetic pigments *etc.* in plants.

Many approaches have been introduced to remove Pb from wastewater. Among these techniques, ion exchange (*Dąbrowski et al., 2004*), electrolysis, electro-coagulation (*Al Hawli, Benamor & Hawari, 2019*), chemical precipitation (*Charerntanyarak, 1999*), solvent extraction, and adsorption (*Iqbal et al., 2021*; *Imran et al., 2020*) *etc.* are commonly used. Moreover, most of these treatment technologies are expensive, generate secondary pollution, require large number of workers and experts, and show less effectiveness (*Izah, Bassey & Ohimain, 2018*). However, adsorption is efficient for the sequestration of HMs from contaminated water (*Imran et al., 2019*).

Metal ions can be removed from wastewater by the adsorption process due to its effective sequestration on the surface of an adsorbent materials (*Tariq et al., 2020*; *Imran et al., 2021*). In this phenomenon, low-cost adsorbents that have higher adsorption potential are utilized. A diversity of indigenously available constituents, *e.g.*, agricultural wastes, natural wastes, and industrial by-products can be employed as economical adsorbents (*Abdel-Salam, 2018*). *Okoye, Ejikeme & Onukwuli (2010)* reported that activated carbon has been extensively employed as an adsorbent for the sequestration of Pb from contaminated aqueous systems. Activated carbon, on the other hand, is still a costly material due to its widespread application in water and wastewater treatment sectors (*Iqbal et al., 2021*)). Therefore, for this purpose, more efficient, economically feasible and environment-friendly adsorbents are required. Several scientists have attempted batch scale experiments to examine the adsorption capacity of orange peels (*Adesanmi et al., 2020*), cocoa shells (*Meunier et al., 2003*), banana peels (*Annadurai, Juang & Lee, 2003*), rice husks (*Elham, Hossein & Mahnoosh, 2010*), saponified melon peels (*Chaudhary & Ijaz, 2014*), eggshell (*Hussain & Shariff, 2014*) and bentonite (*Naseem & Tahir, 2001*) for the elimination of Pb from polluted water. However, literature shows that there are limited studies performed on column scale for the remediation of Pb from contaminated water.

Peanut shells (PS) are thrown away as waste product all over the world. They are permeable and contain (hemi)cellulose material, lignin, pectin and slight amount of protein as well as a very complicated combination of polymeric organic compounds and have high electrolytic ability (*Shah et al., 2018*; *Kocasoy & Güvener, 2009*). These properties can make them effective biosorbent. These shells are plentiful, affordable, easy to process, and incredibly efficient constituents. According to *Chaudhary & Ijaz (2014)*, cost effective adsorbents should be commonly accessible with high selectivity and large capacity for the sequestration of contaminants. Other criteria that is often overlooked is that no or slightly harmful and toxic chemicals are used in the synthesis of these biosorbents. Solid wastes *e.g.*, compost and PS can be used to obtain all these benefits. Furthermore, reuse of these solid wastes for the sequestration of HMs ions can eliminate the issue of disposal of hazardous effluents while also increasing economic benefits (*Tahiruddin & Ab Rahman, 2013*). The goals of present study were to: (i) assess the solid wastes (*i.e.*, peanut shells and commercial compost to remove Pb from wastewater) adsorption potential and (ii) evaluate adsorption of Pb by using these wastes at column and batch scale under variety of adsorbate concentrations, adsorbent dosage, and solution pH and interaction time.

## MATERIALS AND METHODS

### Preparation of adsorbents

The peanut shells (PS) were gathered from the local market in Vehari. The PS were rinsed with distilled water to get rid of the adhered particles and dust. The PS were dried in an oven at 70 °C for 36 h. Subsequently, the PS were grounded and sieved (1 mm mesh). The ground material (peanut shells as biosorbent: PSB) was stored in plastic container for further applications. The commercial compost biosorbent (CCB) was obtained from a

company "PlantFert". The CCB was prepared from vegetables waste as per company instructions.

## Characterization of adsorbents

Representative samples of PSB and CCB were characterized using Fourier transform infrared (FTIR) spectroscopy for functional groups determination (*Imran et al., 2021*). The surface morphology of PSB and CCB was obtained with scanning electron microscopy (SEM) while the adsorbent samples were analyzed with Brunauer Emmett-Teller (BET) to find pore volume and surface area from $N_2$ adsorption/desorption isotherms.

Infrared spectra of pristine PSB and CCB were obtained in absorbance mode from wavenumber 700–4,000 cm$^{-1}$ at resolution of 2 cm$^{-1}$. The FTIR spectra were achieved using a Matson Polaris FTIR spectrophotometer. Prior to FTIR analysis, PSB and CCB were milled with KBr to form fine powder and compressed into thin pellets. The SEM images for surface morphology were obtained using a TESCAN Vega TS 5136LM at 20 kV. The samples for SEM analysis were coated in gold with a Balzers' Spluttering device. The Brunauer-Emmett-Teller (BET) surface area and pores related information of both PSB and CCB were measured with a Tristar 3000 (Micromeritics, Norcross, GA, USA) analyzer. Porosity (%) of the PSB and CCB was measured following *Shah et al. (2019)*.

## Preparation of synthetic wastewater

Lead nitrate (Pb $(NO_3)_2$) salt was used for the preparation of Pb stock solution. For 1,000 mg/L Pb stock solution in a 1,000 mL volumetric flask, 1.60 g of Pb $(NO_3)_2$ was thoroughly mixed in 500 mL distilled water and final volume was made 1 L. Different sub-stocks of 25–100 mg/L were prepared from this stock solution. For the pH adjustment, a few drops of 0.5 M NaOH/HCl solution were added to maintain the pH 6.0. Because dissolved Pb ions precipitate at high pH, maximal adsorption of Pb occurs at 6.0 pH (*Imran et al., 2020*; *Ahmed et al., 2016*).

## Fixed bed column scale removal of Pb and reusability of PSB

The fixed bed column scale experiments for Pb removal were carried out in plexiglass columns. The experiment was conducted at steady state in duplicate sets. The column scale arrangement with the dimensions (14.5 cm length and 4.5 cm internal diameter) is presented in Fig. 1. The adsorbent biomass (3.0 and 6.0 g/column) was packed as a central layer between acid-washed quartz sand (0.1–0.2 mm). Wet packing was adopted to release the trapped air between the particles and to get rid of disturbance during flow of contaminated water in the column. At the inner bottom and topmost position, cloth filter was used to improve the flow distribution, prevent adsorbent loss, and avoid clogging of the openings used for capillaries carrying Pb contaminated water at inlet and effluent at outlet. Filter was also used at the bottom and top of the adsorbent layer as shown in Fig. 1. At the start, distilled water was run for 30 min through the columns to flush the columns. Two different initial concentrations of Pb (25 and 50 mg/L) were injected through the columns at flow rate 2.5 and 5.0 mL/min at pH 6 (optimum pH level at batch scale). At outlet side, effluent was collected in plastic bottles by gravity. The effluent

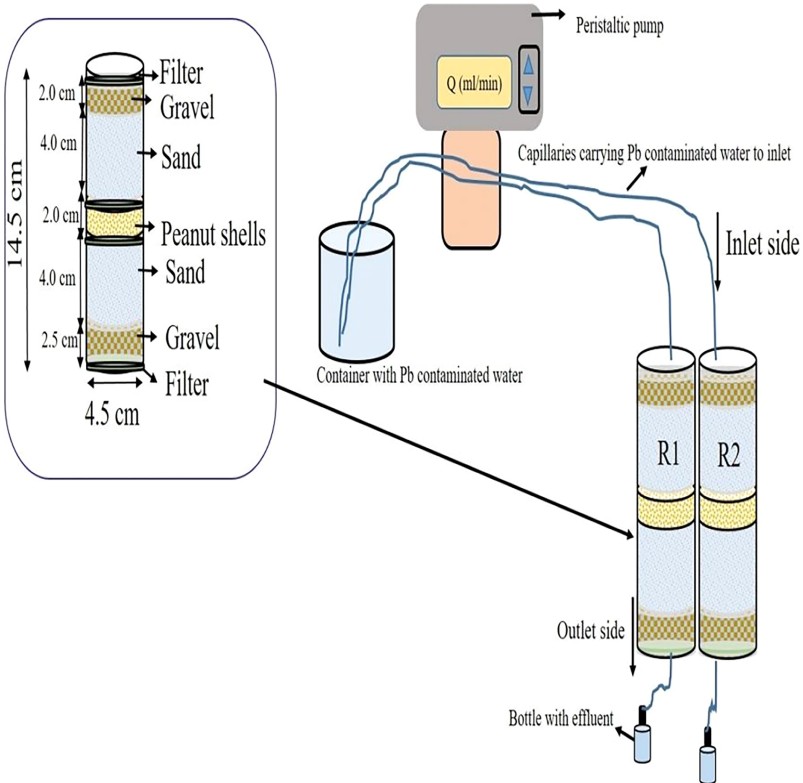

**Figure 1 Schematic representation of column scale arrangement for Pb removal from water.**

was collected after 15, 30, 60, 120, 180, 240, 300, 360, 420, 480, 540 and 600 min to determine the residual Pb concentration in the water after attachment of Pb at PSB and CCB surface. A peristaltic pump was used to inject contaminated water at a uniform flow rate at inlet side while at outside pump was not used just effluent was collected by gravity. In Fig. 1, acronym R1 represents replication one and R2 indicates replication two of the column experimentation. To estimate the reusability of PSB and CCB, the columns were emptied after experimental run at 25 mg/L and the used PSB and CCB were rinsed with 2% HCl solution, filtered by adding distilled water and used for repacking before starting injection of 25 mg/L concentration again in the columns. The column scale removal and reusability of PSB and CCB for Pb sequestration were calculated based on the residual concentration and initial concentration of Pb in water.

## Batch experiments

Batch experiments were performed in the Environmental Sciences laboratory of COMSATS University Islamabad (CUI), Vehari Campus. Digital weighing balance was used to weigh three different adsorbent doses (1.25, 2.5, and 5.0 g/L) of both adsorbents (PSB and CCB). In 250 mL conical flasks, 100 mL of different concentrations (25, 50, 75, and 100 mg/L) of Pb contaminated solution and weighed doses of adsorbents were added. Duplicate sets were prepared and placed on a mechanical shaker at 150 rpm

**Table 1 Characteristics of groundwater samples (GW1 and GW2) used to evaluate the impact of interfering inorganic ions on the removal of Pb (II) from contaminated water.**

| Parameter | Units | GW1 | GW2 |
|-----------|-------|-----|-----|
| EC | dS m$^{-1}$ | 1.85 ± 0.15 | 0.9 ± 0.05 |
| TDS | mg L$^{-1}$ | 850 ± 10.1 | 495 ± 8.1 |
| Mg | mg L$^{-1}$ | 25.5 ± 1.7 | 37 ± 2 |
| Na | mg L$^{-1}$ | 245 ± 5.3 | 60 ± 3 |
| K | mg L$^{-1}$ | 5.1 ± 1.1 | 4 ± 0.2 |
| Ca | mg L$^{-1}$ | 65 ± 2.1 | 30.2 ± 4 |
| CO$_3$ | mg L$^{-1}$ | 80 ± 3.1 | 45 ± 4 |
| HCO$_3$ | mg L$^{-1}$ | 570 ± 7 | 232 ± 8.4 |
| Cl | mg L$^{-1}$ | 690 ± 8.5 | 490.4 ± 8 |
| SO$_4$ | mg L$^{-1}$ | 430 ± 6 | 355 ± 7.5 |
| NO$_3$ | mg L$^{-1}$ | 30.5 ± 3.4 | 45.3 ± 5 |
| Pb (II) | µg L$^{-1}$ | 80 ± 1.6 | 57 ± 3.2 |

and room temperature (28 °C). At different time intervals (15, 30, 60, and 120 min), samples were taken and filtered using Whatman filter papers-42. The filtered samples were used to determine the residual concentration of Pb in water samples. Samples of initial concentrations of sub solution (without addition of biosorbents) were also stored for analysis. For the determination of pH effect on the sequestration of Pb from contaminated water, different pH values (4–10) were adjusted using 0.5 M NaOH/HCl solutions rest while of the parameters *e.g.*, initial concentration (IC), temperature and adsorbent dose were kept constant. Atomic Absorption Spectrophotometer (AAS) was used to measure the concentration of residual Pb in samples.

## Impact of coexisting ions on Pb removal from water

In groundwater and wastewater, there exists several other ions which interfere in the adsorption of contaminant from water onto the adsorbent surface. Therefore, in the present study two groundwater samples (GW1 and GW2) having different concentration of cations and anions collected from Vehari were used to evaluate the impact of coexisting ions in water for the sequestration of Pb. The groundwater samples were analysed for electrical conductivity (EC), total dissolved salts (TDS), pH, sodium (Na), potassium (K), calcium (Ca), magnesium (Mg), carbonate (CO$_3$), bicarbonate (HCO$_3$), Chlorine (Cl), sulphate (SO$_4$), nitrate (NO$_3$) and Pb. The EC, TDS and pH were measured with respective EC, TDS, and pH meters. The Na, K and Ca were measured using Flame photometer while SO$_4$ and NO$_3$ concentration was measured with calorimetric method. The concentration of Cl, CO$_3$ and HCO$_3$ was measured with titration method *Estefan, Sommer & Ryan (2013)* while Mg and Pb were measured using atomic absorption spectrophotometer.

First groundwaters containing Pb (GW1 = 80 ug/L, GW2 = 57 ug/L) were treated with PSB and CCB at optimum conditions and subsequently Pb concentration (25 mg/L) was developed in both groundwater samples and Pb removal was evaluated in the presence

of different cations and anions in water. The characteristics of GW1 and GW2 have been presented in Table 1. The results of batch scale adsorption experiments carried out in distilled water (DW) and groundwater samples at Pb concentration (25 mg/L and 50 mg/L) were compared to evaluate the impact of competing ions in water on Pb removal.

## Data analysis

The initial and final concentrations of Pb ions in water samples were measured using AAS. The equilibrium adsorption $q_e$ (mg/g) of Pb ions onto PSB and CCB was estimated with Eq. (1) (*Imran et al., 2021*; *Shah et al., 2019*; *Edokpayi et al., 2015*).

$$q_e = \left( \frac{C_i - C_e}{W} \right) \times V \tag{1}$$

In Eq. (1), $C_i$ is the IC and $C_e$ is the residual concertation (mg/L) of Pb ions in water, $V$ is the volume of Pb contaminated water (L), W is the mass (g) of the adsorbents (PSB and CCB) and $q_e$ represents the amount of adsorbate (Pb) ions attached on the surface of PSB and CCB at equilibrium. To determine the kinetic adsorption $q_t$ at time $t$, $Ce$ in Eq. (1) was replaced with residual concentration ($C_t$) at time $t$.

The percentage removal (R%) of Pb ions from contaminated water by PSB and CCB at column and batch scale was found from initial and final concentration ($C_f$) of Pb using Eq. (2)

$$R(\%) = \left( \frac{C_i - C_f}{C_i} \right) \times 100 \tag{2}$$

The total volume of Pb containing solution ($V_t$) injected to each column was calculated from the multiplication of flow rate ($Q$) and total injection time $t_{total}$ ($V_t = Qt_{total}$). The total mass ($m_{total}$) of metal ions (mg) injected to the column was found using Eq. (3) while column scale adsorption was estimated from Eq. (1) by replacing $V$ with $Qt_{total}$.

$$m_{total} = \frac{C_i Q t_{total}}{1000} \tag{3}$$

## Adsorption kinetics modeling

Kinetic studies are employed for the optimization of different operating conditions for the adosorption process. Literature shows that different kinetic models are proposed to explain the reaction sequence and adsorption behavior (*Imran et al., 2020*; *Iqbal et al., 2021*). In the present study, pseudo first order (PFO), pseudo-second order (PSO), and intra-particle diffusion (IPD) kinetic models were used for the kinetics of Pb ions adsorption onto PSB and CCB. The correlation coefficients ($R^2$) were used to determine the applicability of these kinetic models. The model is most relevant to data when the $R^2$ value is high (close to 1.0) (*Imran et al., 2019*).

The PFO kinetic model is based on the notion that the change in contaminant concentration overtime is proportional to power one which has been presented in linear form (Eq. 4).

**Table 2 Parameters of kinetic and equilibrium adsorption models.**

| Type | Model | Parameter | PSB | CCB |
|---|---|---|---|---|
| Equilibrium adsorption models | Langmuir | $K_L$ (L/mg) | 0.080 | 0.077 |
| | | $q_{max}$ (mg/g) | 42.5 | 26.39 |
| | | $R^2$ (–) | 0.89 | 0.99 |
| | Freundlich | $K_F$ ($[mg/g(L/mg)^{1/n}]$) | 5.36 | 3.96 |
| | | $n$ (–) | 1.92 | 2.27 |
| | | $R^2$ (–) | 0.98 | 0.99 |
| Kinetic models | Pseudo first order | $k_1$ (1/min) $\times 10^{-2}$ | 0.046 | 0.069 |
| | | $q_e$ (mg/g) | 9.58 | 5.47 |
| | | $R^2$ (–) | 0.889 | 0.92 |
| | Pseudo second order | $k_2$ (g/mg/min) $\times 10^{-2}$ | 0.0075 | 0.0028 |
| | | $q_e$ (mg/g) | 16.7 | 15.6 |
| | | $R^2$ (–) | 0.99 | 0.99 |
| | Intra-particle diffusion | $k_{dif}$ | 0.51 | 0.83 |
| | | C | 9.85 | 4.11 |
| | | $R^2$ (–) | 0.67 | 0.77 |

$$\log(q_e - q_t) = log q_e + \frac{k_t}{2.303} t \qquad (4)$$

In Eq. (4), $q_e$ and $q_t$ represent the adsorption capacity of PSB and CCB (mg g$^{-1}$) at equilibrium and time $t$, respectively, $k_1$ represents rate constant (L min$^{-1}$) and $t$ represents time duration (min). Table 2 shows the values of $k_1$, $q_e$, and R$^2$ which is a correlation coefficient between experimental and model values. The findings indicated that a plot of log($q_e$–$q_t$) vs $t$ using the Lagergren PFO model yields a straight line with a low correlation coefficient (R$^2$). In comparison to the experimental results, the PFO kinetic model projected much lower values of the equilibrium adsorption capacity ($q_e$). The results showed that the PFO kinetic model is incompatible with the kinetic data of Pb adsorption. The PFO kinetic model is not usually appropriate for entire data range of interaction time, but it explains well the adsorption mechanism at the start of the interaction between the adsorbent and contaminant (*Ho & McKay, 1999*).

The PSO kinetic model explains the contaminant adsorption mechanism throughout the whole interaction time. The PSO may be expressed in linear form by Eq. (5):

$$\frac{t}{q_t} = \frac{1}{k_2 q_e^2} + \frac{1}{q_e} t \qquad (5)$$

where $k_2$ (g mg$^{-1}$ min$^{-1}$) indicates the PSO rate constant of the contaminant adsorption process. The values of $k_2$ (g mg$^{-1}$ min$^{-1}$) and $q_e$ (mg g$^{-1}$) were determined by making a plot of $t/q_t$ against $t$. The values of PSO model parameters and R$^2$ for adsorption of Pb have been presented in Table 2. The estimated and experimental $q_e$ values were found to be closer to each other with high R$^2$. Thereafter, PSO kinetic model fits well with the kinetic adsorption of Pb ions on PSB and CCB as compared with PFO kinetic model. These findings corroborate the previously published findings (*Sadaf & Bhatti, 2011*). Several

stages are involved in the attachment of Pb molecules from contaminated water to the surface of adsorbent materials. The rate-controlling step in a batch experiment system includes quick and continuous stirring might be film diffusion, intra-particle diffusion (Eq. 6), or a combination of both processes.

$$q_t = K_{dif}t^{0.5} + C \tag{6}$$

In Eq. (6), C = intercept which characterizes boundary layer thickness and $k_{dif}$ (mg g$^{-1}$ min$^{-1/2}$) = IPD rate constant. Table 2 shows the $k_{dif}$ and C values for Pb adsorption onto PSB and CCB. According to IPD hypothesis, curve between $q_t$ and $t^{0.5}$ should be linear. If IPD is involved in the adsorption of a contaminant, there exists straight line between a plot of contaminant adsorbed per unit mass of biosorbent ($q_t$) and $t^{0.5}$ (Gupta & Bhattacharyya, 2006). The low value of R$^2$ suggests that the Pb ion attachment onto the PSB and CCB is not influenced IPD.

## Adsorption equilibrium isotherms

The adsorption isotherms are employed to test the adsorbents adsorption capacity for a certain contaminant (Imran et al., 2020). The adsorption isotherm is a property of a system at a given temperature (Iqbal et al., 2021; Shah et al., 2019; Poulopoulos & Inglezakis, 2006). Adsorption isotherms may be predicted using a variety of models. Langmuir and Freundlich equilibrium sorption isotherms are the most widely utilized models for pollutant adsorption from wastewater (Tariq et al., 2020).

In Langmuir isotherm (Eq. 7), monolayer adsorption of a contaminant onto the surface with a limited number of sites available for contaminant adsorption is considered.

$$q_e = \frac{q_{max}K_LC_e}{1 + K_LC_e} \tag{7}$$

where $q_{max}$ represents the maximum adsorption of Pb ions onto PSB and CCB, $K_L$ = Langmuir model constant and $C_e$ = Pb concentration in solution (mg/L) at equilibrium.

Langmuir isotherm model in linearized form has been given in Eq. (8) to find the model parameters. The values of the Langmuir model parameters, $q_{max}$ and $K_L$ were found by using the slope and intercept of the linear plot of $C_e/q_e$ vs $C_e$.

$$\frac{C_e}{q_e} = \frac{1}{q_{max}K_L} + \frac{1}{q_{max}C_e} \tag{8}$$

The biosorbent surface might be monolayer or/and multilayer. It is reported that Freundlich isotherm model (Eq. 9) is applicable for multilayer adsorption and it assumes contaminant interaction with a heterogeneous surface having a non-uniform sorption energy distribution throughout the surface.

$$q_e = K_fC_e^{1/n} \tag{9}$$

where $q_e$ = equilibrium sorption of Pb per mass of the PSB and CCB (mg/g), $C_e$ = aqueous concentration of Pb at equilibrium (mg/L), $K_f$ and $n$ are Freundlich model constants related to the adsorption capacity and the adsorption intensity, respectively.

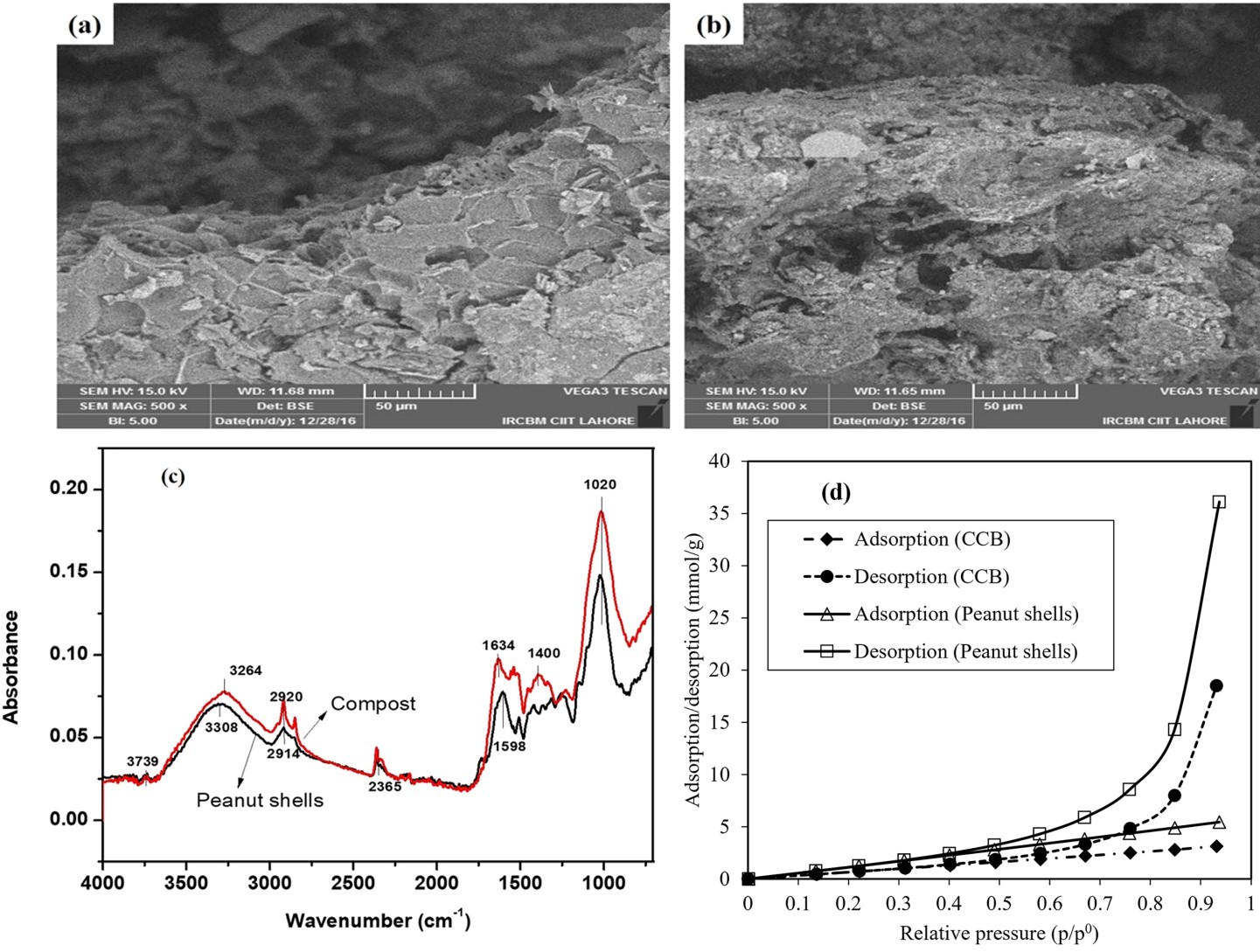

**Figure 2** SEM image of Peanut shells (A), CCB (B) and (C) FTIR spectra of both adsorbents, (D) N$_2$ adsorption/desorption during BET analysis of both CCB and PSB used for the removal of Pb from water at fixed bed column scale and batch scale.

To determine the values of model parameters, linearized form of Freundlich isotherm (Eq. 10) model was used. The slope and intercept of the curve of ln $q_e$ vs ln $C_e$ were considered to derive the Freundlich equilibrium constants $K_f$ and $n$.

$$\ln q_e = \ln K_f + \frac{1}{n}\ln C_e \qquad (10)$$

# RESULTS AND DISCUSSION

## Characterization of adsorbents

The surface morphology of PSB and CCB using scanning electron microscope has been presented in Figs. 2A and 2B. The SEM results illustrated that PSB and CCB have porous surface with pores having variation in size and possessed irregular shapes containing

capillary tubes that might help to trap metal ions through diffusion. Overall, there is more aggregation of particles in case of CCB as compared with PSB which might lead to lower Pb adsorption by CCB.

Figure 2C displays the Fourier transform infrared (FTIR) spectroscopic results to find the functional groups existing on PSB and CCB surface. As shown in Fig. 2C, the FTIR spectra indicated broad band at 3,308 cm$^{-1}$ which represented –OH groups. The absorbance peak observed at wavenumber 2,920–2,849 cm$^{-1}$ indicates aliphatic C–H group. The one peak observed at 1,400 cm$^{-1}$ represents alkanes (C–H) bending. The absorption peaks observed at wavenumber 1,518–1,634 cm$^{-1}$ represent stretching of C=C. The presence of these functional groups on PSB and CCB surface is responsible for the Pb adsorption from contaminated water. However, there is difference in absorbance peak and wavenumber between PSB and CCB which might help in their different Pb adsorption potential.

Brunauer Emmet-Teller (BET) analysis was performed to find the surface area and pore size of PSB and CCB adsorbents. The N$_2$-adsorption/desorption isotherms obtained from BET analysis were used to determine the pores related information (size, volume, porosity) and BET surface area of PSB and CCB (Fig. 2D). There is difference in adsorption/desorption curves which reveals the porous nature of both adsorbents. The BET results revealed that PSB exhibits surface area of 226.8 m$^2$/g. The average particle and pore size of PSB was 26.5 and 3.3 nm, respectively. While PSB has 5.11% porosity which contributes in the electrostatic attachment and diffusive movement of Pb onto PSB surface. The surface area of CCB was 126.1 m$^2$/g with particle size of 45 nm and pore size of 3.5 nm.

## Breakthrough curves for Pb removal at fixed bed column

It was important to assess the impact of controlling parameters on the removal of Pb from fixed bed column with both adsorbents. The column scale removal of contaminants is governed by flow rate, inlet concentration, bed depth and inner diameter of the column (Canteli et al., 2014). In the present study, the impact of flow rate (2.5 and 5.0 mL/L), initial concentration of Pb (25 and 50 mg/L) and dose (bed height = 0.65 and 1.2 cm/column) of PSB and CCB (3.0 and 6.0 g/column) was evaluated on the shape of breakthrough curves.

## Impact of flow rate on column scale Pb sequestration

Figure 3A presents the breakthrough curves for the removal of Pb onto PSB and CCB at two different flow rates (2.5 and 5 mL/min) while other parameters; dose of PSB and CCB was 3.0 g/column (0.65 cm/column) were kept constant constant. The results revealed that when flow rate is increased, there is decline in the uptake of Pb ions by both adsorbents (PSB and CCB) due to negative effect of flow rate on mass transfer efficiency. The volume of injected Pb containing water at 2.5 and 5.0 mL/min was 1,500 mL and 3,000 mL, respectively. While bed volume (BV) number which is the ratio of total

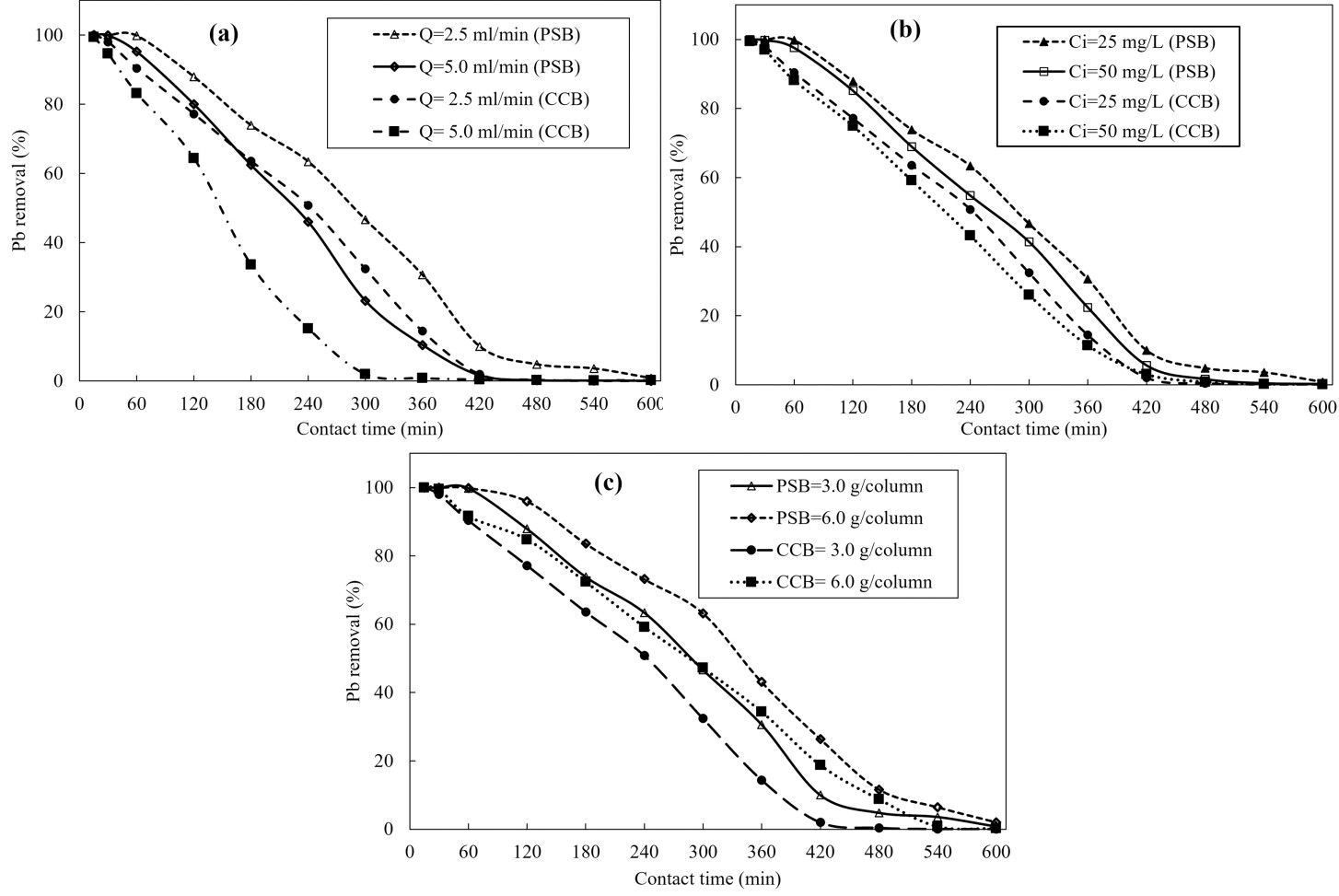

**Figure 3 Breakthrough curves obtained from fixed bed column scale removal of Pb using Peanut shells (PSB) and commercial compost biosorbent (CCB); (A) effect of flow rate (2.5 and 5.0 mL/min) when dose of PSB and CCB is 3.0 g/column pH-6 and Ci = 25 mg/L, (B) impact of initial concentrations of Pb (25 and 50 mg/L) on its removal, (C) impact of biomass of PSB and CCB when flow rate is 2.5 ml/min and pH-6.**

volume of contaminated water fed to the column and volume of adsorbent bed was 157. The results show that volume of treated effluent is reduced at higher flow rate because breakthrough curve is obtained earlier, and adsorbents get saturated in short time at high flow rate of injected water. It is attributed to the low interaction time between the adsorbent material and Pb ions which ultimately prevents complete saturation of the active sites thereby leading a decline in adsorption of Pb ions. Similar trend was reported by other investigators in literature (*Naeem et al., 2019*; *Cantelli et al., 2014*; Caprine et al., 2013). The results show that Pb ions adsorption on PSB strongly depends on the flow rate at which contaminated water is being injected to the column. The PSB and CCB takes more time to get saturated when there is low flow rate and sorption increased steadily as compared with injection of Pb containing water at high flow rate. The injection time must be extended when contaminant is injected at low flow rate because adsorption sites

are saturated gradually (*Naeem et al., 2019*). Overall, PSB showed more removal of Pb ions as compared with CCB. At the start, there is negligible difference in Pb removal by PSB and CCB (first 30 min) while later on *e.g.* at 300 min, PSB removed 47% Pb and CCB removed 32.4% Pb from contaminated water at constant flow rate (2.5 mL/min). However, results showed that at the same (300 min) PSB and CCB showed just 23.7% and 2.5% removal, respectively when at flow rate 5.0 mL/min. Similarly, *Yahya et al. (2020)* studied that the decline in metal ions onto the cashew nutshell adsorbent was observed when flow rate was increased at a fixed adsorbent bed. It is due to the less time available between adsorbents (PSB and CCB) and metal ions for interaction which reduces the adsorbent saturation. While at lower flow rate metal ions were removed efficiently. *Xu, Cai & Pan (2013)* reported that at lower flow rate film resistance and thickness was increased which contributes to higher metal removal from contaminated water. *Swapna Priya & Radha (2016)* reported that the adsorption was significantly affected by varying flow rate. As flow rate increases the film resistance is decreased which reduces the sequestration of contaminants from aqueous systems.

## Effect of initial concentration of Pb on column scale sequestration

Figure 3B displays the effect of Pb initial concentration (25 and 50 mg/L) on its removal by PSB and CCB at column scale when flow rate is 2.5 mL/min, bed height is 0.65 cm (3.0 g/column) and pH of Pb containing water is pH-6. The mass of Pb fed to the adsorbents was 37.5 mg and 150 mg when water was injected at 25 and 50 mg/L, respectively. The results show that the change in concentration gradient has significant impact on the saturation of PSB and CCB and their Pb removal (%) from contaminated water. There is decline in Pb removal when its concentration was changed from 25–50 mg/L. This decline in Pb removal at elevated concentration of Pb (50 mg/L) is accredited to the saturation of available sorption sites on PSB and CCB beyond a certain initial level of metal ions. At time 240 min, PSB and CCB showed 63.4% and 50.8% Pb removal, respectively when initial Pb concentration was 25 mg/L while these adsorbents showed 54.8% and 43.2% Pb removal, respectively at the same time (240 min) but at 50 mg/L initial Pb concentration. However, Pb adsorption onto PSB and CCB increased with increase in initial concentration of Pb in water. This enhanced Pb adsorption at elevated initial Pb level is because of higher utilization of active sites at elevated Pb concentration. This earlier exhaustion of the PSB and CBB active sites at elevated Pb level is attributed to the fact that high Pb concentration caused quick saturation of the sorbent. Results (Fig. 3B) reveal that the shape of curves by PSB and CCB at higher concentration (50 mg/L) is more pronounced and steepness of the slope is also increased. *Swapna Priya & Radha (2016)* reported that the adsorption capacity decreased from 10.26 to 9.90 μg g$^{-1}$ when the solution initial concentration decreased from 600 to 200 μg L$^{-1}$. However, there is slower saturation of adsorbent materials at lower initial concentration as compared with high initial concentration. Similarly, *Bharathi & Ramesh (2013)* reported the removal of contaminant decreased with increasing initial concentration which is due to decline in exhaustion and breakthrough time. *Dorado et al. (2014)* reported that the breakthrough time is decreased at enhanced metal concentration.

## Effect of dose (bed height) on column scale Pb removal

Figure 3C presents the impact of bed height of PSB and CCB filled in the column for Pb removal at steady state (constant flow rate) when column was wet packed with 3.0 g and 6.0 g PSB and CCB separately. The impact of bed height was evaluated at constant initial Pb concentration (25 mg/L) and flow rate 2.5 mL/min. The results revealed that when dose was changed from 3.0 to 6.0 g/column, there was improvement in Pb removal at a given time which is attributed to higher number of adsorption sites and more height of bed for the attachment of Pb at higher dosage (6.0 g/column which is equal to 1.2 cm bed height) as compared with 3.0 g/column which is equal to 0.65 cm/column. The PSB and CCB showed 63.4% and 50.8% Pb removal, respectively at dose 3.0 g/column while at higher dose (6.0 g/column), PSB and CCB showed higher removal (73.2% and 59.3%, respectively). The increase in bed height results in more service time for breakthrough curve and exhaustion time for a given concentration (Naeem et al., 2019; Imran et al., 2021; Canteli et al., 2014). Bharathi & Ramesh (2013) reported that the removal increased with increasing bed height/dose. It is attributed to more adsorption sites available for the contaminant sequestration. El Mouhri et al. (2020) observed that the removal enhanced from 95.4% to 97.4% when the bed height was increased from 3 to 5 cm. The increase in removal percentage is attributed to the active sites and interaction time available for the contaminant with the adsorbent material. Swapna Priya & Radha (2016) studied that the quantity of adsorbent is very crucial for the breakthrough point because adsorbent quantity determines the bed height. As bed height is increased, the breakthrough would also be higher because increase in bed height increases the surface area significantly which favors the adsorption.

## Batch scale Pb sequestration from contaminated water
### Impact of initial Pb level on its adsorption and removal

The impact of various concentrations of Pb with constant adsorbent mass on Pb removal (%) are shown in Fig. 4A. Figure 4A depicted that the Pb removal decreased with the increasing initial Pb concentration in contaminated water. For PSB, a sharp decrease was observed from 90.0% to 79.8% when the solution initial concentration was changed from 25 to 50 mg/L. While for CCB under same conditions, a decline in Pb removal from 79.2-52.5% was observed. It was attributed to the limited number of active available sites for the removal of Pb from contaminated water. As the solution initial concentration increased, there is more competition between adsorbent and adsorbate. At lower initial concentration, the removal was highest because PSB contains enough available spaces that favors the adsorption of Pb on PSB and CCB. While at higher concentration, the available spaces become saturated with Pb ions and no more sites are available for the adsorption of Pb ions. The decline in the Pb removal with increasing solution concentration is attributed to the concentration gradient or split in the flux in solute concentration and sorbent surface. The decline in removal with increasing concentration was due to the decline in available sorption sites (Boudrahem et al., 2019). Alatabe & Al-Sharify (2019) reported that the rapid adsorption of Pb at lower concentration. While with the increasing concentration of Pb the adsorption decreased, which is due to the Pb

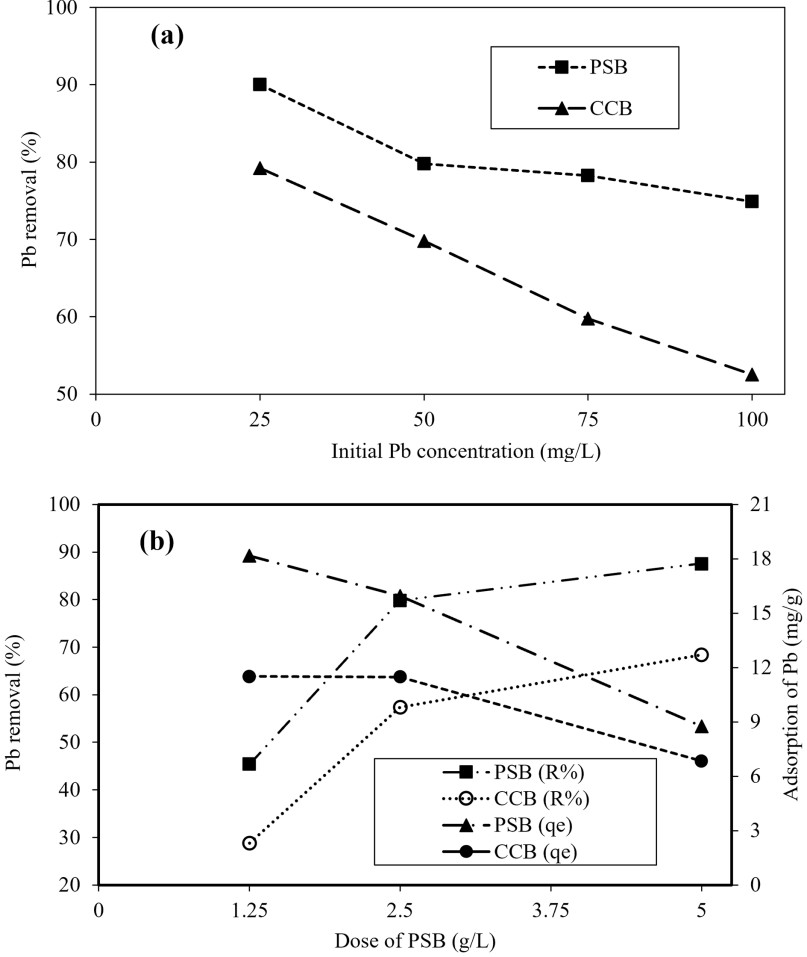

**Figure 4** **(A) Effect of initial concentration of Pb (25–100 mg/L) on its removal (%) at equilibrium, pH-6, dose 0.5 g/L and room temperature; (B) impact of PSB and CCB dose on Pb removal at equilibrium, Ci = 50 mg/L, pH-6.**

ions adsorbed on the outer surface of the adsorbent. Furthermore, enhanced initial solution concentration of Pb resulted into fast saturation of sorbent materials.

### Impact of adsorbent dose on Pb removal

Figure 4B depicts that effect of PSB and CCB adsorbents dose for the removal of Pb from contaminated water by varying adsorbent dose from 1.25 to 5.0 g/L of contaminated water when Pb initial concertation was 50 mg/L. Results revealed a clear impact of PSB and CCB biomass on Pb removal (Fig. 4B). The maximum values of Pb ion sorbed on PSB and CCB surface were found at mass of 1.25 g/L of contaminated water. Overall, Pb adsorption showed a decline with the increase when adsorbent dose was increased. As adsorbent dose is increased, surface area also increases and the interaction between Pb and adsorbent becomes strong therefore, the removal increased with increasing adsorbent dosage. Initially, a sharp increase (45.4% to 79.8%) in Pb removal was observed By PSB when dosage was changed from 1.25 to 2.5 g/L of Pb contaminated water. While with the further increase in adsorbent dosage the impact was small (79.8–87.6%) which is due to the

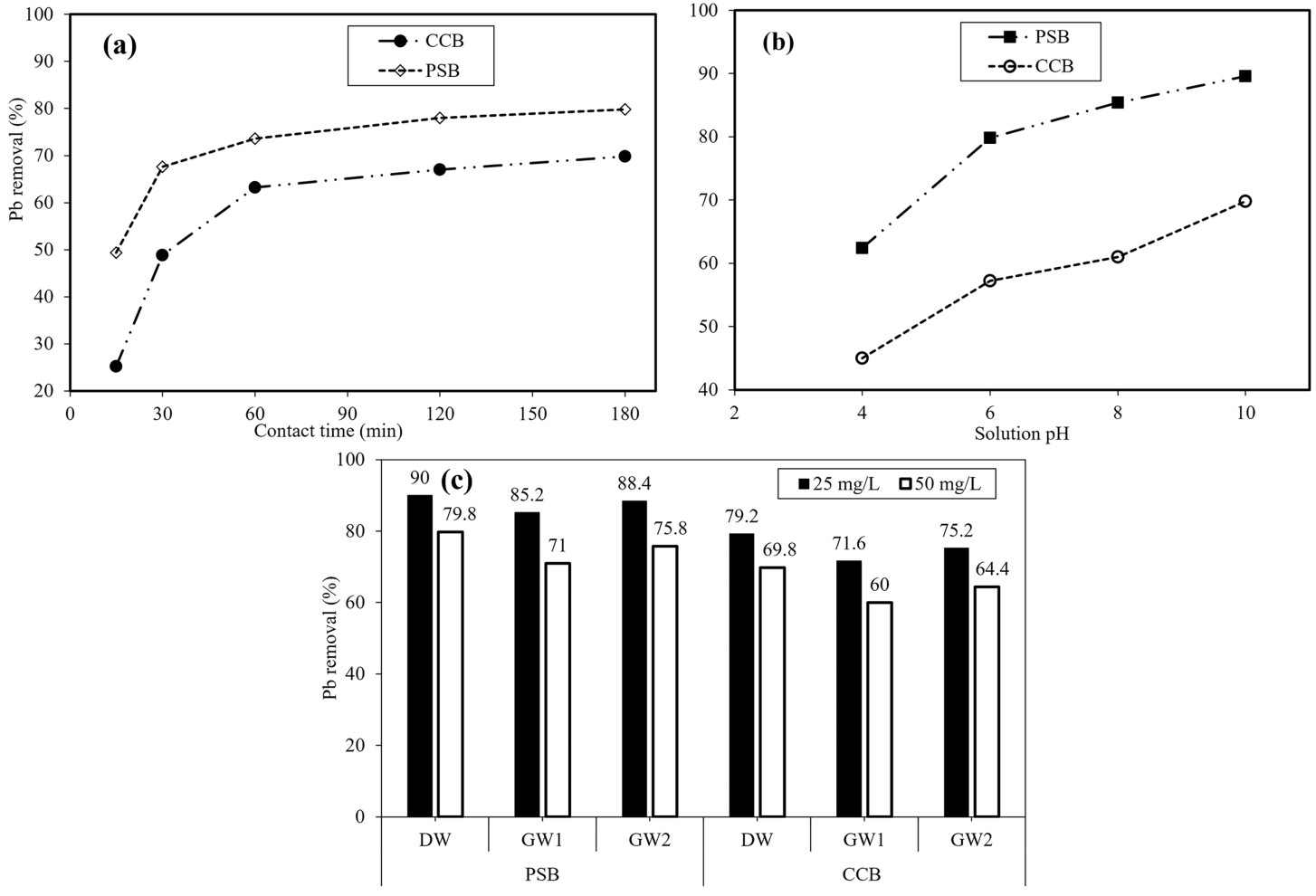

**Figure 5** Effect of interaction time on, (A) Pb removal when Ci = 50 ppm, PSB dose = 0.5 g/100 mL), (B) impact of solution pH of contaminated water on Pb removal at equilibrium, dose 0.5 g/100 mL, Ci = 50 mg/L, (C) impact of competing ions in groundwater sample on Pb removal at 25 and 50 mg/L, and pH-6.

possible aggregation of adsorbent particles at higher dose (*Imran et al., 2021*). Similarly, *Iqbal et al. (2021)* reported that with the addition of adsorbent dosage the removal was elevated which is attributed to the competition between adsorbent and adsorbate. Initially there are limited number of active available sites and higher number of contaminant ions therefore, the removal was decreased. Moreover, at lower adsorbent dosage the active sites becomes saturated with contaminant earlier (*Imran et al., 2020*). At higher adsorbent dosage, the removal was not significantly increased which is due to the overlapping/ aggregation of adsorbent material at higher concentration of dose (*Wang et al., 2018*). Similar results were reported by many researchers (*Adenuga et al., 2019*; *Heraldy et al., 2018*; *Iqbal et al., 2021*; *Kostić et al., 2014*).

### Effect of interaction time on Pb removal

Effective adsorption is reflected when the equilibrium is attained with adsorbent and adsorbate. Furthermore, a certain time is crucial that confirms that the equilibrium has

been attained (*Dakhil, 2015*). Therefore, impact of interaction time was evaluated as a main parameter that affects the adsorption capacity of PSB and CCB adsorbents. Figure 5A displays the behavior of Pb removal with different time interval on PSB and CCB at constant pH-6, dose 2.5 g/L, and concentration 50 mg/L. Adsorption of Pb increased when interaction time was increased but a significant changes happened in first 30 min, subsequently minor variations in adsorption were noted. The adsorption of Pb ions became constant when at equilibrium (60 min) by both PSB and CCB adsorbents. At equilibrium, highest Pb ions removal efficiency was 79.8% when 50 mg Pb/L interacted with PSB and CCB showed 48.80% removal under same conditions. The increase in removal with increasing interaction time is attributed to the interaction time between adsorbent and Pb ions. Initially the adsorption sharply increased but with the passage of time, the Pb removal was slower than initial period of time. After 60 min, change in removal is low due to occupation of most of the adsorption sites during first 60 min. The removal onto PSB increased from 49.4% to 73.0% when the interaction time was changed from 15 to 60 min. Results show that when interaction time further increased from 60 to 180 min, only 6.8% more Pb was removed. Therefore, experiment was run for 180 min to attain equilibrium. *Mukherjee et al. (2020)* reported that the Pb and Cd removal gradually increased with increasing interaction time and then a sharp decline was noticed after 30 min, which is due to the intra-particle diffusion process. Similarly, *Gaur et al. (2018)* reported that the equilibrium was attained in first 60 min for the adsorption of Pb and As by using soya been biosorbent. The results were also consistent with other relevant studies (*Arimurti, Heraldy & Lestari, 2016*; *Ben-Ali et al., 2017*; *Yogeshwaran & Priya, 2021*).

## Effect of solution pH on Pb removal

In adsorption studies, pH plays an important role for the removal of contaminants from wastewater. The impact of pH for the removal Pb onto PSB and CCB from contaminated water was evaluated. Therefore, the solution pH was changed from 4 to 10, while all other parameters were kept constant (50 mg/L). Figure 5B demonstrates the impact of pH on Pb removal from contaminated water. It was noticed that with the increase in pH, the removal percentage also increased. The removal percentage with PSB changed from 62.0–79.9% when the solution pH was increased from 4 to 6. The increase in removal percentage is due to fluctuations in pH which might result from the surface properties of the adsorbent material as well as contaminant molecules. *Heraldy et al. (2018)* reported that the removal enhanced with the increasing pH of the Pb contaminated solution. Previously reported that the effectiveness of the adsorbent materials were low at lower pH and increased with increasing pH (*Pang et al., 2011*). This might be due to the fact that lower pH can charge the active sites with H+ ions and makes the adsorbent surface more positive that cause electrostatic repulsion between adsorbent and Pb ions. Moreover, at basic pH electrostatic attraction occurred between adsorbent and adsorbate and enhanced removal was obtained (*Hafshejani et al., 2015*; *Taşar, Kaya & Özer, 2014*). When the solution pH > 6, the Pb hydrolysis and precipitation occurs as Pb hydroxides (*Heraldy et al., 2018*). As pH was further changed from 6 to 10, there is improvement of Pb removal

but removal at pH beyond 6 is attributed to Pb precipitation in solution as we observed clear precipitates at this higher pH (8 and 10). So, pH-6 was selected as optimum pH and all other cases were conducted at this pH.

## Impact of coexisting ions on Pb removal

The presence of competing ions in water may affect the contaminant removal onto the adsorbent surface (*Imran et al., 2019*; *Vilvanathan & Shanthakumar, 2015*). Figure 5C presents the results regarding the impact of cations and anions on the removal of Pb when initial Pb concentrations (25 and 50 mg/L) were developed in two groundwater samples (GW1 and GW2). The characteristics of GW1 and GW2 are given in Table 1. The results (Fig, 5C) reveal that Pb removal with DW, GW1 and GW2 onto PSB at 25 mg/L was 90%, 85.2% and 88.4%, respectively while PSB showed 79.8%, 71.0% and 75.8% Pb removal in DW, GW1 and GW2, respectively at 50 mg/L. The GW1 and GW2 showed 4.8% and 1.6% lower Pb removal than DW when experiment was run at 25 mg/L while GW1 and GW2 showed 8.8% and 4.0% less Pb removal as compared with DW at 50 mg/L. Similarly, Pb removal with DW, GW1 and GW2 onto CCB at 25 mg/L was 70.2, 71.6 and 75.42, respectively while CCB showed 69.8%, 60.0% and 64.4% Pb removal in DW, GW1 and GW2, respectively at 50 mg/L. The GW1 and GW2 showed 7.6% and 4.0% lower Pb removal than DW when experiment was run at 25 mg/L while GW1 and GW2 showed 9.8% and 5.4% less Pb removal as compared with DW at 50 mg/L. The GW1 showed more difference in Pb removal onto both PSB and CCB as compared with GW2. This higher difference in Pb removal by GW2 is attributed to higher concentration of cations (Na, K, Ca, and Mg) as compared with GW1 (Table 1). The higher concentration of coexisting cations (especially Ca and Mg) competed for adsorption sites with Pb ions (*Imran et al., 2019*; *Meseguer et al. 2016*) to occupy the surface of PSB and CCB adsorbents. Similar response of competing ions in groundwater was found by *Imran et al. (2019)* on the removal of Pb from contaminated water. Likewise, many other studies have reported that with increasing ionic strength the removal decreased (*Imran et al., 2020*, *2021*; *Kariuki, Kiptoo & Onyancha, 2017*).

## Equilibrium and kinetic isotherms for Pb adsorption onto PSB

The equilibrium data for Pb adsorption by PSB and CCB was mathematically explained by adsorption isotherm models as described earlier in Methodology section. The kinetic adsorption data obtained at 50 mg Pb/L was validated with PFO, PSO and Intra-particle diffusion model. The fitting behavior of kinetic models has been displayed in Figs. 6A–6C. The comparison of modeling results with kinetic experimental data shows that pseudo second order well explained the kinetic adsorption of Pb onto PSB and CCB as is evident from the value of coefficient of determination ($R^2$) which is 0.99 for Pseudo second order model. The values of kinetic and equilibrium model parameters have been shown in Table 1. For equilibrium data, values of coefficient of determination show that Pb ion sorption well fitted with Freundlich adsorption isotherm ($R^2 = 0.98$, Table 1). The fitting behavior of equilibrium models (Langmuir and Freundlich) has been presented in

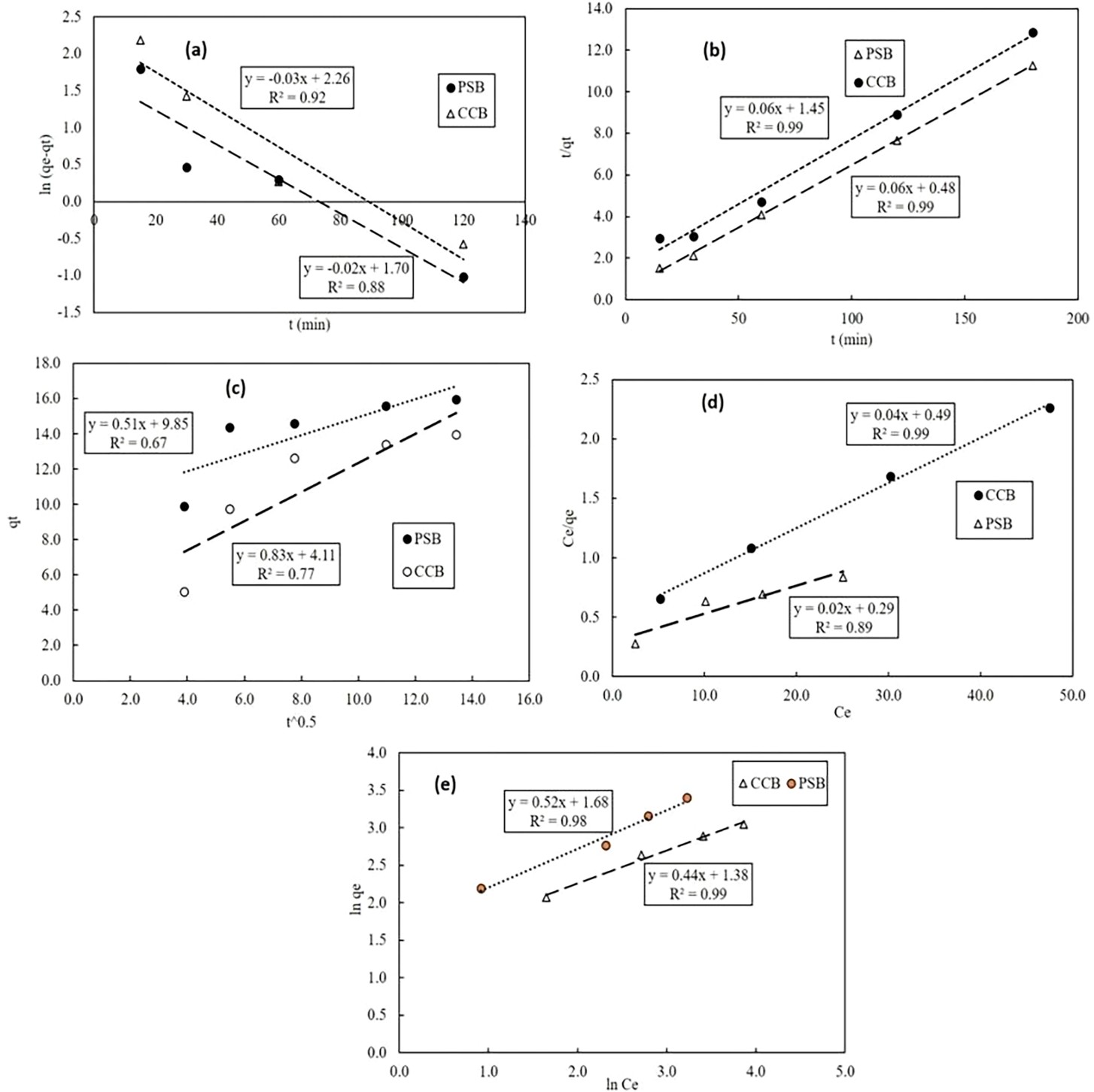

**Figure 6 Correlation of experimental data with kinetic adsorption models: Pseudo first order (A), pseudo second order (B) and intra-particle diffusion (C); equilibrium adsorption isotherms: Langmuir (D), Freundlich (E).**

Figs. 6D–6E. The respective values of coefficient of determination $R^2$ in case of Langmuir adsorption isotherm fitted with PSB was 0.89 while adsorption by CCB was equally fitted with Langmuir model as its $R^2 = 0.98$. The value of $q_{max}$ for PSB and CCB as per

**Table 3 Comparison of different adsorbents for the removal of Pb from contaminated water.**

| No. | Materials | Adsorption/Removal | References |
|-----|-----------|--------------------|-----------|
| 1 | lignin biosorbent | 89.02 mg/g | *Klapiszewski et al. (2017)* |
| 2 | soya bean | 0.72 mg/g | *Gaur et al. (2018)* |
| 3 | Pineapple waste | 77.16 mg/g | *Mopoung & Kengkhetkit (2016)* |
| 4 | Melon peel | 72 mg/g | *Gour et al. (2016)* |
| 5 | Palm fruit fiber | 73 mg/g | *Ideriah, David & Ogbonna (2012)* |
| 6 | metallothionein | 39.02 mg/g | *Mwandira et al. (2020)* |
| 7 | Modified corncobs | 4.34 mg/g | *Mendoza-Castillo et al. (2015)* |
| 8 | Okra waste | 5.00 mg/g | *Hashem (2007)* |
| 9 | *Agave sisalana* | 1.34 mg/g | *Dos Santosa et al. (2011)* |
| 10 | Tea waste | 1.35 mg/g | *Mondal (2010)* |
| 11 | Eupatorium adenophorum spreng | 3.46 mg/g | *Guo et al. (2009)* |
| 12 | Peanut shells biosorbent | 50.0 mg/g | Current study |

Langmuir model was 42.05 mg/g and 26.39 mg/g, respectively. Moreover, respective $n$ values of Freundlich model for PSB and CCB was 1.92 and 2.27.

## Comparison of Pb removal by PSB with literature

A comparison of Pb adsorption onto PSB and CCB with previous studies is important to determine the effectiveness and real time applications of proposed materials. Table 3 highlights the comparison of PSB and CCB with previous studies. Current study depicted that the adsorption of Pb was 50.0 mg/g. Hence, it was noticed that the proposed material (PSB) is more efficient than other adsorbents such as corncobs (*Mendoza-Castillo et al., 2015*), soya bean (*Gaur et al., 2018*), Eupatorium adenophorum spreng (*Guo et al., 2009*), Okra waste (*Hashem, 2007*), Tea waste (*Mondal, 2010*), and *Agave sisalana* (*Dos Santosa et al., 2011*). It was also noticed that many other adsorbents are available that have been more efficient such as tomato waste and apple juice residue (*Heraldy et al., 2018*), lignin biosorbent (*Klapiszewski et al., 2017*), Pineapple waste (*Mopoung & Kengkhetkit, 2016*), Palm fruit fiber (*Ideriah, David & Ogbonna, 2012*) *etc.* than PSB. However, the availability and cost effectiveness of the adsorbent materials is also very important. Although many adsorbents are more efficient, but the process and cost effectiveness makes PSB attractive for the remediation of Pb from contaminated water.

## CONCLUSIONS

In the present work, the fixed bed column and batch scale adsorption potential of Peanut shells and commercial compost as biosorbents (PSB and CCB) were successfully evaluated for the sequestration of Pb from contaminated work. The maximum adsorption of Pb onto PSB (42.5 mg/g) was higher than CCB (26.39 mg/g) at optimum dose and pH, which is better than several previously explored adsorbents. The more Pb sequestration onto PSB is attributed to its improved surface characterization (SEM, FTIR, and BET) as compared with CCB under same conditions. The presence of competing ions in groundwater (GW1 and GW2) showed 9.8% and 5.4% lower, Pb removal by CCB as

compared with distilled water while this difference was 8.8% and 4.0% by PSB when 50 mg Pb/L was used under optimal conditions. The kinetic behavior of Pb adsorption onto the adsorbents showed best fit with pseudo second order kinetic model ($R^2 = 0.99$) while isothermal equilibrium adsorption onto PSB was best explained with Freundlich model ($R^2 = 0.98$), however, Pb adsorption onto CCB was equally well explained with Langmuir and Freundlich adsorption models. Column experiments showed excellent retention of Pb especially on PSB under different flow rates, initial Pb concentration and dose of the adsorbents. It is concluded that PSB is more effective and environment friendly at both column and batch scale for the removal of Pb as compared with CCB.

## ACKNOWLEDGEMENTS

Authors are thankful to COMSATS University Islamabad (CUI), Vehari Campus to providing research facility and technical expertise. We are equally indebted to IRCBM, CUI Lahore-Campus for the characterization of the biosorbents.

### Funding

The authors did not receive funding for this work.

### Competing Interests

The authors declare that they have no competing interests.

### Author Contributions

- Ghulam Mustafa Shah conceived and designed the experiments, authored or reviewed drafts of the paper, and approved the final draft.
- Muhammad Imran conceived and designed the experiments, analyzed the data, prepared figures and/or tables, and approved the final draft.
- Umme Aiman performed the experiments, prepared figures and/or tables, and approved the final draft.
- Muhammad Mohsin Iqbal performed the computation work, authored or reviewed drafts of the paper, and approved the final draft.
- Muhammad Akram analyzed the data, performed the computation work, authored or reviewed drafts of the paper, and approved the final draft.
- Hafiz Muhammad Rashad Javeed conceived and designed the experiments, prepared figures and/or tables, and approved the final draft.
- Atika Waqar performed the computation work, authored or reviewed drafts of the paper, and approved the final draft.
- Faiz Rabbani analyzed the data, prepared figures and/or tables, and approved the final draft.

### Data Availability

  The raw data is available in the Supplemental Files.

## Supplemental Information

Supplemental information for this article can be found online at http://dx.doi.org/10.7717/peerj-pchem.21#supplemental-information.

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
