# Peer review of "Efficient sequestration of lead from aqueous systems by peanut shells and compost: evidence from fixed bed column and batch scale studies"

_PeerJ Physical Chemistry, doi:10.7717/peerj-pchem.21_

## Round 0.1 · original submission · Major Revisions

The basic idea of your experiment is acceptable and interesting but the development of your job is questionable, as fully explained in the comments of the reviewers. Your paper is not acceptable in its present form so it should be revised; you need further experimental work and revised analysis as explained in detail in the comment of the reviewers, especially the first and the third reviewers. The problem of using only distilled water is a strong limitation.

Read with attention the critical comments of the reviewers and be ready to perform further experiments if you want to publish your idea. Regards.

Reviewer 1 ·

Basic reporting

This submission deals with the adsorption of lead from water onto a biosorbent based on peanut shells. Some equilibrium batch tests were conducted and conventional models were applied to the data. The approach applied is very conventional and the only new aspect is the use of a relatively new biosorbent for a specific contaminant. I think that it is a normal approach in a laboratory to evaluate a commercial product and to propose it for the treatment of a specific type of water. There are many studies and textbooks giving detailed research and insights about the experimental evaluation of adsorbents. I don’t see any advancement in the science in this aspects.
It can further not be excluded that such a waste product may have the potential to contaminate the treated water because of pesticides or other heavy metals used in various part of the world.
Because of its low informative value and no innovative approaches proposed, I have to recommend to reject the submission.

Experimental design

- Line 125: in this research a synthetic wastewater was used, based on a very simple composition by the dilution Pb in distilled water. This is one of the main concerns in this research, because the use of synthetic wastewater does not permit to evaluate the effects of non-competitive and competitive adsorption and desorption (mutual interference effects, multiple adsorption mechanisms, ...).
- Please reconsider the extensive use of the words "biosorption" - they refer to filling materials able to develop a microbial community or biofilm.

Validity of the findings

- Figure 4: It is very questionable to apply these models on 3-4 data points.
- A very important aspect which is not covered by the authors is that distilled water was examined. If, as the authors suggest, the tested material should be useful for contaminant removal from wastewaters etc. why were no tests carried out on such waters? This would be highly necessary because in real/ natural waters, competition for adsorption sites occurs, reducing the available adsorption capacities for single adsorbates by orders of magnitude.
- With the extremely high dosages reported here (for instance see figure 4 with 25-100 mg/L), it appears largely unrealistic that the tested material could be applied in practice.

Additional comments

no comment

·

Basic reporting

The paper is written in an English that needs improvement. It's not bad, but there are some mistakes that need to be corrected. Just one example: line 468 <<The more Pb sequestration onto PSB is attributed to its improved surface characterization>>. Some words are spelled wrong (e.g., line 177 concertation instead of concentration).
Literature references look fine, and sufficient field background/context is provided.
The article structure, figures, and tables are fine, but a few things should be corrected:
• Figure 3: the caption is incomplete: (b) effect
• Figure 3a: The experimental data symbols for Q = 5.0 ml/min (CCB) are missing - there is only the curve.
• Figure 5a: the line connecting the experimental points must be made either broken (straight lines) or curved: not a mix of the two options.

Experimental design

The experimental design is fine, even if basic. The scientific and methodological approaches are fine. Methods are well described.

Validity of the findings

Findings are OK, although often quite trivial.
I do not see any statistics on the experimental data: it seems all the tests have been performed only once. However, considering the type of tests carried out, I think this can be accepted.
Conclusions seem fine.

Additional comments

Abstract: Indicating the pump flow rate has no physical meaning if the value of other parameters is not reported: bed volume and/or vessel cross-section.
Line 64: define the CNS acronym.
Line 110: if possible, report about the residual biomasses used to produce the commercial compost.
Line 134: specify what do you mean with steady state – I guess the fluid-dynamics.
Line 144: avoid @
Line 177: the IC acronym should be defined.
Line 203 << Table 1 shows the values of k1, qe (calculated from model), qe-experimental,>>. In Table 1 I do not see any experimental data.
Line 225 <<Kp>>: in Table 1 the authors wrote <<Kdif>> referring to the same variable: decide which symbol to use. The IPD acronym should be defined.
Line 326 <<capacity decreased from 10.26 to 9.9015μg g−1>> too many values after the comma in the second data.
Line 374 << The maximum values of Pb ion sorbed on PSB and CCB surface were found at mass of 1.25 g/L of contaminated water. Overall, Pb adsorption showed a decline with the increase when adsorbent dose was increased.>>. Looking at the Fig, the trend seems reversed.
Line 447: << for PSB and CCB was 2.27 and 1.92>>. The values were reversed.

·

Basic reporting

The manuscript deals with the very serious problem of aqueous systems contamination by lead, a very toxic and hazardous contaminant. The solution investigated to obtain pure and safe water for drinking is based on the adsorption performance of a peanuts shell biosorbent.
In the latest years many papers, aiming at finding an efficient, economically feasible and environment-friendly adsorbent, have been published, the most of them addressed to claim very high adsorption capacity. Less attention was addressed to verify the performance of such adsorbents in flow systems based on adsorption packed columns for contaminated water remediation.
The authors claim that main goal of the study was to assess the biosorption potential of peanut shells wastes (PSB) to remove Pb from wastewater through experimental and modeling approaches, using batch and flow systems, operated under various conditions in terms of adsorbate concentration, adsorbent dosage, solution pH and contact time. A comparison with a commercial compost biosorbent (CSB) was also performed.

Experimental design

Experimental characterization of PSB has been carried out through FTIR, SEM, and nitrogen adsorption for surface area/pore volume size.
Really, the results obtained in the latter case (surface area of 226.8 m2 /g, average particle and pore size 26.5 and 3.3 nm, respectively, 5.11% porosity) are not presented as adsorption isotherms, while they are essential to support the reported, rather surprising, size values.

Validity of the findings

The work is medium quality, and could be improved by adding some results, discussion, especially concerning the potential application, following the below suggestions.
Concerning the adsorption equilibrium measurements and modeling:
line 404…… “The removal onto PSB increased from 49.4 to 73.0 % when the contact time was changed from 15 to 60 min. Moreover, only 6.8% removal was increased when the contact time further increased from 60 to 180 min”
Comments : If the purpose is to determine the real equilibrium values, further 6.8 % increase of adsorbed amount is not negligible. The approach to the equilibrium can be very slow, especially in the experiments in which the amount of lead is not enough excess with respect the amount of adsorbent, resulting in a rather high reduction of lead concentration and then driving force to adsorption. Moreover, I suggest that the Authors don’t use the term contact time along all the manuscript to indicate the run time. The modeling approach is acceptable; the same for choosing R2= 0.99 as acceptable values for supporting a model.
Concerning the flow column experiments and modeling :
I believe an introduction to the comments is necessary. The challenge of using adsorption to remove Pb from wastewater containing 10-100 ppm and over is really very much arduous, if one considers that the current limit for drinking water is 10 ppb. This means that you need a very large amount of very high capacity adsorbent or a very large scale adsorption column. Cheap adsorbent can surely help, but how much the adsorbent cost influences the final cost of the treatment ? And, looking at the above premise, how frequent is the necessity of column regeneration ? How much the operation influences the final cost, compared to the effect of adsorbent cost ? Moreover, the discussion and interpretation of the breakthrough curve sas depending of the operating parameters should be less qualitative.
Coming back to the right comment of the Authors on the low attention addressed in the most papers to verify the performance of adsorbents in flow systems, however their experimental approach is rather far from answering the above considerations. First of all, the scale of the experiments : the column is too small and the adsorbent bed geometry is questionable. A too large diameter with respect the height also results in a radial profile of velocity very non uniform, and the entrance effects are critical.
From the experimental data it can be seen that to reach the limit of 50 ppb (higher than the current regulation limit) 99.8 or 99.9 % removal are the values fot the breakthrough curves to be stopped. One simple parameter to be evaluated is the corresponding BV (bed volume) number at that value, i.e. the ratio between the total amount of liquid fed to the bed and the adsorbent volume. Few BVs means a very high frequency of regeneration, resulting in high operation cost.
One more aspect, the importance of regeneration, underlined by the Authors, is not investigated. The comparison with CCB is winning in terms of adsorption capacity and shape of breakthrough curves, however the flow column behavior of CCB is very poor. It seems difficult to imagine the practical utilization of CCB for lead sequestration in a flow system on the base of the results found by the Authors.
English language is of acceptable quality.

---

## Round 0.2 · accepted · Accept

The paper is now acceptable

·

Basic reporting

It seems to me that the paper has been improved by introducing further experimental tests (as requested by other reviewers) and accepting the observations / suggestions / requests for changes highlighted by me in the first round of review.

Experimental design

It seems to me that the paper has been improved by introducing further experimental tests (as requested by other reviewers) and accepting the observations / suggestions / requests for changes highlighted by me in the first round of review.

Validity of the findings

It seems to me that the paper has been improved by introducing further experimental tests (as requested by other reviewers) and accepting the observations / suggestions / requests for changes highlighted by me in the first round of review.